# Advanced practice providers versus medical residents as leaders of rapid response teams: A 12-month retrospective analysis

**Herman G. Kreeftenberg**[1,2]☯*, **Ashley J. R. De Bie**[1,2,3]☯, **Eveline H. J. Mestrom**[2,3]☯, **Alexander J. G. H. Bindels**[1,2]☯, **Peter H. J. van der Voort**[4,5]☯

**1** Dept Internal Medicine, Catharina Hospital Eindhoven, Eindhoven, Netherlands, **2** Dept of Intensive Care, Catharina Hospital Eindhoven, Eindhoven, Netherlands, **3** Dept of Electrical Engineering, Eindhoven University of Technology, Eindhoven, Netherlands, **4** TIAS School for Business and Society, Tilburg University, Tilburg, Netherlands, **5** Dept of Critical Care, University Medical Centre Groningen, Groningen, Netherlands

☯ These authors contributed equally to this work.
* herman.kreeftenberg@catharinaziekenhuis.nl

**Data Availability Statement:** All relevant data are within the paper and its Supporting Information files.

## Abstract

### Purpose

In a time of worldwide physician shortages, the advanced practice providers (APPs) might be a good alternative for physicians as the leaders of a rapid response team. This retrospective analysis aimed to establish whether the performance of APP-led rapid response teams is comparable to the performance of rapid response teams led by a medical resident of the ICU.

### Material and methods

In a retrospective single-center cohort study, the electronic medical record of a tertiary hospital was queried during a 12-months period to identify patients who had been visited by our rapid response team. Patient- and process-related outcomes of interventions of rapid response teams led by an APP were compared with those of teams led by a medical resident using various parameters, including the MAELOR tool, which measures the performance of a rapid response team.

### Results

In total, 179 responses of the APP-led teams were analyzed, versus 275 responses of the teams led by a resident. Per APP, twice as many calls were handled than per resident. Interventions of teams led by APPs, and residents did not differ in number of admissions (p = 0.87), mortality (p = 0.8), early warning scores (p = 0.2) or MAELOR tool triggering (p = 0.19). Both groups scored equally on time to admission (p = 0.67) or time until any performed intervention.

### Conclusion

This retrospective analysis showed that the quality of APP-led rapid response teams was similar to the quality of teams led by a resident. These findings need to be confirmed by prospective studies with balanced outcome parameters.

**Funding:** The author(s) received no specific funding for this work.

**Competing interests:** The authors have declared that no competing interests exist

## Introduction

Hospital medicine is dealing with patients with increasingly complex disorders that require a highly efficient and high-quality healthcare organization [1, 2]. Rapid response systems with teams led by physicians have been shown to reduce in-hospital cardiopulmonary arrests and mortality [3, 4]. However, the organization of these rapid response systems is subject to the worldwide emerging shortages of physicians, especially in rural areas [5, 6].

These shortages force numerous hospitals to reorganize their rapid response systems and other teams in order to be able to continue to provide a 24/7 coverage.

One option that has been considered is that a rapid response team might be led by different health care professionals, ranging from attending physicians to nurses. Limited scientific evidence suggests that teams led by a physician perform better than teams led by non-physicians [7, 8]. In practice however, an increasing proportion of in-hospital acute and emergency care is delivered by junior clinicians in the first years of their training, including the responsibility of leading a rapid response team, which might reduce the efficacy and quality of these teams. One of the potential solutions is to reallocate this responsibility to physician assistants or nurse practitioners, also called advanced practice providers (APP). This profession is gaining recognition in critical care which is supported by clinicians who recognize their quality and continuity of their work [9, 10].

Very few experiences with APPs as leaders of a rapid response team have been reported [11, 12].

Two previous studies provided some guidance about the outcomes of patients visited by a team led by an advanced practice provider, but inter-comparability is hampered by differences between the considered health care systems and by the lack of validated outcome parameters. A third, retrospective single center study comparing outcome data of rapid response teams led by a nurse practitioner and by a registrar showed an improved hospital mortality in the nurse practitioner -led group after propensity matching. This study mainly reported patient outcomes [13].

The main objective of the present study was, to establish whether the performance of APP-led rapid response teams is comparable to the performance of teams led by a medical resident of the ICU, focusing on process- as well as patient-related outcomes.

## Methods

### Study design and setting

We performed a single-center retrospective cohort study over a period of 12 months. This time period was chosen to reduce the influence of confounders, such as changes within the organization of the hospital and the ward, for instance the implementation of a completely new operational Electronic Medical Record (EMR) in the hospital or the use of continuous monitoring devices within certain departments. The study was performed in the Catharina Hospital Eindhoven, a Dutch tertiary teaching hospital which houses all specialties except for neurosurgery and transplantation surgery. The hospital has a 33-bedded ICU, which facilitates a mix of post-operative cardiac and oncologic surgery and, on the other hand, specialties such as, neurology, pulmonology and gynecology and internal medicine, including dialysis. During the period of the study, clinical protocols regarding the rapid response system remained unchanged.

### Patient selection

The EMR was queried to identify patients who had been assessed by the rapid response team. Patients and assessments were eligible if a bed-side assessment had been performed by the

rapid response team in patients aged 18 years of older. Medical consultations by telephone were excluded.

## Rapid response team triggering, modified early warning score and the MAELOR tool

On the wards, the modified early warning score is used as described by v Galen et al. [14] to identify deteriorating patients or patients in need of advanced care. In short, this tool assigns points to abnormal physiological parameters and in turn triggers a rapid response team call. It also provides the opportunity to call in support if there is a sense of unease about the condition of the patient.

The MAELOR tool is a validated tool to measure and quantify the performance of a rapid response team [15]. This tool consists of a flow chart which is triggered if the patient has a high modified early warning score and stops triggering if a patient is admitted within 4 hours after the initial call or has a resolution of critical clinical symptoms within 48 hours. The tool also stops triggering if there are treatment limitations and if ICU admittance is not warranted.

The MAELOR tool flow diagram is depicted in Fig 1. Clinical variables necessary for the MAELOR tool were only recorded until 48 hours after the initial call.

## Rapid response team organization

The organization of the local rapid response system has been described in the COMET study, a multicenter study that evaluated the implementation of structured rapid response systems in the Netherlands [3]. Since implementation of the rapid response team in the Catharina Hospital Eindhoven in 2013, These teams have consisted of an ICU nurse and a team leader, the leader being either a medical resident working on the ICU or an advanced practice provider. One of the medical residents or advanced practice providers on the ICU manages the pager for the rapid response team during duty. A call to the rapid response team can be made if a patient on the ward scores a modified early warning score of $\geq 3$ points or if a nurse experienced a substantiated sense of worry about a patient. This call can either be made by medical residents or by registered nurses on the general ward. Usually, when the modified early warning score indicates a critically ill patient, the nurse on the ward informs the resident, who in turn decides if a rapid response team assessment is necessary. The rapid response team call is postponed if the cause of the high early warning score is known and additional treatment, such as an operation, has been planned or if a sepsis can be treated on the normal ward.

The team carries a basic set of materials, which consists of resuscitation fluids, masks for supplemental oxygen and lifesaving medication, such as glucose or phenylephrine.

Following the first assessment, the leader of the rapid response team discusses the case with the intensivist on call for the ICU. This consultation results either in ICU admission or in recommendations for treatment on the general ward, which can include changing treatment limitations, such as the "do not resuscitate" (DNR) code or the "do not intubate" (DNI) code and other treatment limitations. If it was decided that the patient is to be admitted to the ICU, the rapid response team transports the patient and admits him or her to the ICU. After the admission to the ICU the leader of the rapid response team remains responsible for the care of the patient until the end of his duty, together with a dedicated ICU nurse. Shortly after ICU admission, the intensivist visits the patient in the ICU. The hospital has a separate team for non-ICU-related in-hospital cardiac arrests. In case of a cardiac arrest the rapid response team is involved if the patient experiences a return of spontaneous circulation and will be admitted to the ICU or the cardiac care unit (CCU).

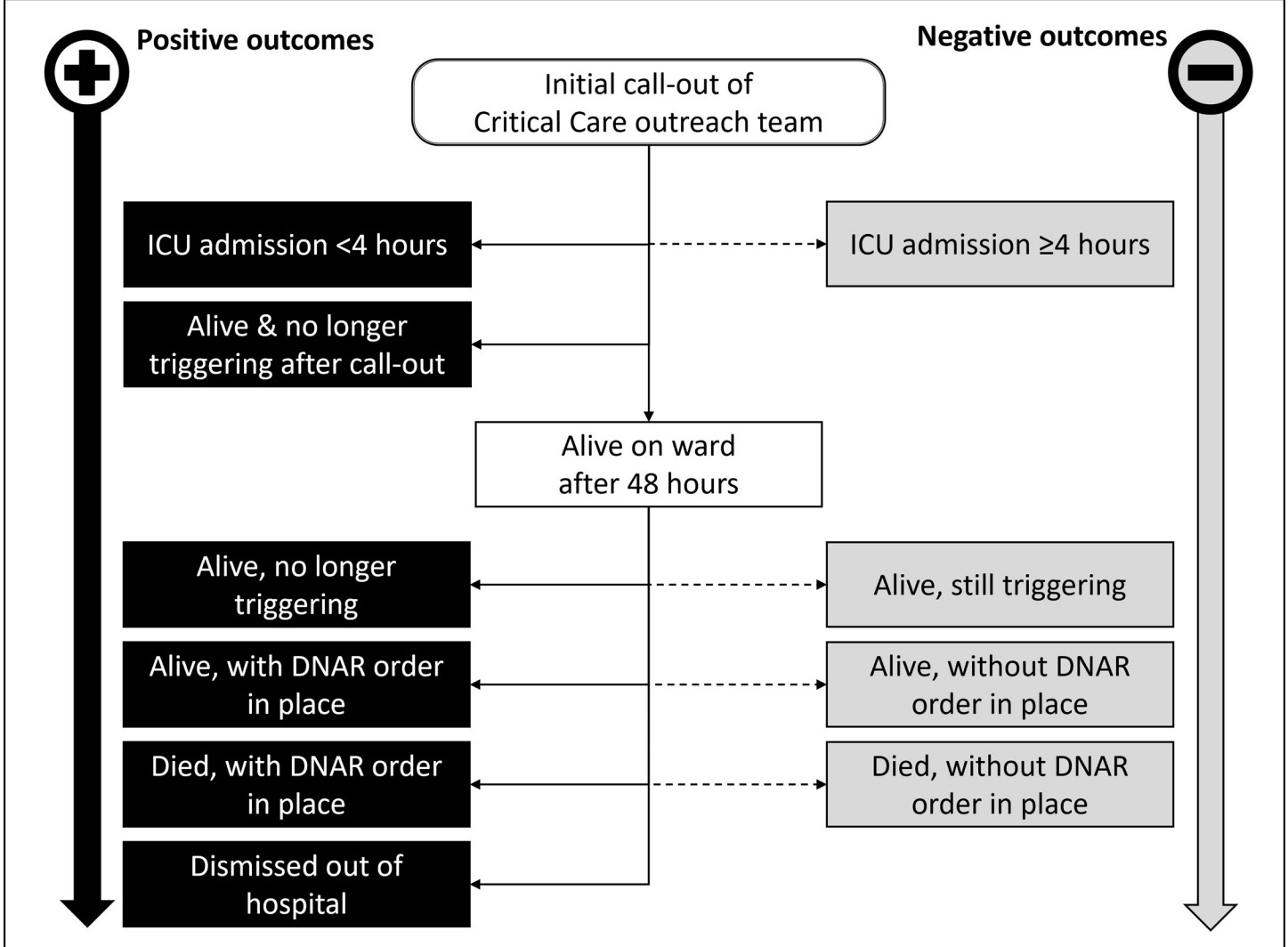

**Fig 1. Flow chart MAELOR tool.**

### Advanced practice providers and medical residents

The advanced practice providers who work in the ICU are qualified as physician assistants. They received a 2.5-year training in the medical domain, which grants them a master's degree, and after graduation, they are qualified to perform all ICU tasks autonomously. The APPs all worked as ICU-nurse before their training to become an APP. They work in collaboration with intensivists. The medical residents originate from the following disciplines: internal medicine, cardiology, pulmonology, and surgery. The medical residents attend an internship during a period ranging from 3 months to 1 year.

### Shifts

The ICU ward uses a system of rotating shifts with a minimum of four clinicians in each shift (advanced practice providers and medical residents). Usually, six of these clinicians are present during the day shift. During the evening shift, two or three of these clinicians are present, and

during the night shift, two. The number of FTE in the entire group of advanced practice providers is 4.99, and in the MR group it is 10.69.

## Ethics

Approval for the study was obtained from the national and regional Ethics committee in accordance with Dutch and European legislation (Medical Research Ethics Committees United (MEC-U); W17.095). A local applicability permission was obtained separately. This article was prepared using the Standards for Quality Improvement Reporting Excellence (SQUIRE) guidelines [16].

## Data

Data of all the consecutive rapid response team visits over a period of 12 consecutive months (2017–2018) were retrospectively extracted from the hospital data base. The patient variables collected were age sex, diagnosis, hospital admission, discharge data, death, Apache IV score on admission and after 24 hours, all blood samples before and after the rapid response team consultations and treatment limitations. Additionally, the composition of the team was noted.

Since there are no mandatory rules for the composition and organization of a rapid response team, the performance and efficiency of different teams are difficult to measure and compare. We gathered several parameters to measure the efficiency of the process. The parameters were categorized in three groups. First, the patients' outcome data: length of stay, mortality, and if applicable, treatment limitations. Second, parameters of team performance: to measure team performance, we used the time until change in early warning system score together with the time until various interventions: the time between the consultation of the rapid response team and the arrival on the ICU, the time between the consultation and interventions such as central or arterial catheterization or intubation. Arterial and central venous lines can be inserted both by residents and by APPs. Intubations by residents are performed under supervision. Third, the MAELOR tool, a validated instrument to assess the performance of rapid response teams was scored.

## Statistics

The data were analyzed with SPSS statistical package version 25 (IBM corporation, Armonk, NY, USA). Means are reported with standard deviations for normal distributions and medians with interquartile range are reported for other distributions. Parameters which were recorded once every hour were considered continuous. Categorical independent variables were compared using the Chi-square test with Yates continuity correction. Categorical and continuous independent non-parametric variables were compared with the Mann Whitney U test, and for the independent parametric variables were analyzed with the independent samples t-test.

Categorical variables with two continuous variables at different points were compared using the mixed between-within subjects analysis of variance was used. A p-value of $p < 0.05$ was considered statistically significant.

## Results

All 454 consecutive rapid response team calls during the assessed period were included in the analysis. Because not all patients received every treatment that was assessed in this study, data on antibiotic change, central venous access, arterial catheters and intubation were not available for all patients.

The team was led by an advanced practice provider in 179 cases and by a medical resident in 275 cases. Of the 454 rapid response team calls, 296 resulted in the patient being admitted to the ICU. This represents approximately 10% of the total yearly ICU admittances. The percentages of rapid response calls resulting in an admission to the ICU were comparable between teams led by an APP and those led by a medical resident (118 (65%) vs 178 (66%), p = 0.78). The level of experience of the APPs was a median of 6.25 years (3.33y-8.25y). In general, an APP handled twice as many calls as a MR.

Table 1 presents the baseline characteristics of patients assessed by APP-led teams and by teams led by a medical resident. Most patients were assessed in the emergency department and on the internal medicine ward. The APACHE IV score of the patients indicates a high severity of illness. No significant differences were found between these two groups except for diastolic blood pressure, which was significantly higher in patients assessed by MR-led teams.

Both the patient- and the process-related outcomes are described in Table 2.

The baseline characteristics of the patients assessed by teams led by advanced practice providers and medical residents demonstrate that no statistically significant differences were encountered between the groups except for diastolic blood pressure. The differences in blood pressure were very small and are therefore considered clinically unimportant.

Concerning the validated MAELOR tool, we were able to retrieve the MAELOR tool outcome in 451 of the 454 cases. Three cases were excluded due to insufficient data. In the analysis of the MAELOR tool outcomes, no significant differences were found between patients treated by teams led by an advanced practice provider or by a medical resident.

Since the rapid response team leader in our ICU model remains responsible for the care provided to the admitted ICU patients, the efficiency of the team could be reflected in the time until arterial line insertion, the time until central venous catheter insertion and the time until

**Table 1. Patient characteristics on arrival of the rapid response team.** Data are given as numbers with percentages or as medians with IQR.

| | Leader of rapid response team | | p-value |
|---|---|---|---|
| | APP Median (IQR) n = 179 | MR Median (IQR) n = 275 | |
| Age (years) | 68 (56–76) | 70 (58–78) | 0.19 |
| Sex (male) | 99 (55%) | 168 (61%) | 0.26 |
| Sex (female) | 80 (45%) | 107 (39%) | 0.26 |
| Apache IV predicted mortality | 58 (42–86) | 62 (37–76) | 0.89 |
| Temperature (degrees Celsius) | 37.3 (36.9–38.5) | 37.4 (37.0–38.4) | 0.80 |
| Systolic Blood Pressure (mmHg) | 120 (99–140) | 128 (109–151) | 0.06 |
| Diastolic Blood Pressure (mmHg) | 69 (50–75) | 70 (60–80) | 0.03 |
| Pulse (rate/min) | 108 (86–124) | 100 (85–119) | 0.28 |
| Respiratory Rate (rate/min) | 25 (18–30) | 20 (16–30) | 0.37 |
| *Location of outreach for ICU-admitted patients* | *APP Number (%)* | *MR Number (%)* | |
| Surgery | 18 (6.1%) | 26 (8.8%) | 1.00 |
| Internal Medicine | 21 (7.1%) | 24 (8.1%) | 0.40 |
| Cardiology | 4 (1.4%) | 6 (2.0%) | NA |
| Pulmonology | 14 (4.7%) | 24 (8.1%) | 0.82 |
| Cardiothoracic surgery | 7 (5.9%) | 2 (1.1%) | NA |
| Neurology | 2 (0.7%) | 8 (2.7%) | NA |
| Gastroenterology | 7 (2.4%) | 7 (2.4%) | 0.61 |
| Emergency department | 38 (12.8%) | 73 (65.8%) | 0.16 |
| Other | 7 (5.9%) | 8 (4.5%) | NA |

APP: advanced practice provider, MR: medical resident, IQR: interquartile range

intubation. These variables did not differ significantly between patients attended to by a team led by an APP or by a resident (Table 2).

To determine the efficiency of non-technical procedures, we also compared the times until change of administered antibiotics in the first 12 hours after admission to the ICU. In the 296 patients admitted to the ICU, the time until change of antibiotics after ICU admission was not significantly associated with the rapid response team leader being an APP or a resident (Table 2). In addition, no association was found between the leader of the rapid response team being an APP or a medical resident and the time until antibiotics administration after ICU admission.

The ICU LOS was determined for 283 of the 296 ICU admitted patients. No significant difference in ICU LOS was found between the patients attended to by APP-led teams and those attended to by rapid response teams led by a medical residents.

The early warning score was assessed at two different points in time (on admission and after 24h). There was no significant difference between the rapid response team leader and the early warning score, or the reduction in early warning score after 24 hours.

Of the 158 patients who remained on the general ward after the rapid response team visit, we were able to extract the early warning score after 24 hours for 123 patients. The warning scores after 24 hours and the reduction in warning scores did not differ between the APP-led teams and the resident-led teams.

In 126 out of 452 patients new treatment limitations were applied after the rapid response team visit. There was no significant association between the instigation of treatment limitations and the rapid response team leader (Table 2). In addition, no effect of rapid response

**Table 2. Outcome variables.**

| | Leader of rapid response team | | p-value |
|---|---|---|---|
| | Response by APP: N(%), Med (IQR) | Response by MR: N(%), Med (IQR) | |
| Number of calls | 179 | 275 | |
| Admission ICU | 118 (66%) | 178 (65%) | 0.87 |
| Time to ICU (hours) | 1.19 (0.56–1.75) | 1.16 (0.59–1.75) | 0.67 |
| *Within 24h*: | | | |
| Hospital mortality | 13 (7%) | 17 (6%) | 0.80 |
| ICU mortality | 6 (5%) | 10 (6%) | 1.00 |
| Time to insertion arterial line (hours) | 1.68 (0.87–2.94) | 1.54 (0.78–2.72) | 0.50 |
| Time from visit to insertion Central venous catheter(hours) | 2.17 (1.24–5.78) | 1.71 (0.92–3.30) | 0.30 |
| Time from visit to intubation (hours) | 3 (1.5–16) | 2 (1.07–10.50) | 0.24 |
| MAELOR not triggering anymore (good outcome) | 165 (92%) | 261 (96%) | 0.19 |
| MEWS admission | 4.04 (2.03–6.29) | 3.92 (1.98–6.56) | 0.90 |
| MEWS at 24 hours | 2.13 (1.04–3.46) | 1.63 (0.51–3.06) | 0.12 |
| Δ MEWS between leaders | | | 0.20 |
| Change in antibiotics | 27 (30%) | 40 (29%) | 1.00 |
| Time to change of antibiotics (hours) | 1.33 (0.62–2.25) | 1.40 (0.65–2.30) | 0.77 |
| ICU LOS (days) | 1.00 (0.20–2.79) | 1.10 (0.17–3.43) | 0.80 |
| Treatment limitation initiated (%) | 54 (30%) | 72 (26%) | 0.44 |

MEWS: Modified Early Warning Score, APP: advanced practice provider, MR: medical resident, IQR: interquartile range

team leadership on mortality was not found in the patients who were deceased on the ward or in the ICU within 24 hours after the RRT visit.

## Discussion

The present study provides insight into the performance of APPs as leaders of rapid response teams in direct comparison with medical residents. In this retrospective study, we found no differences in either process-related or patient-related outcomes between rapid response teams led by APPs and teams led by medical residents. This comparability included the trend of the early warning score after the call and the triggering of the MAELOR tool 48 hours after the call, a tool validated for assessing the quality of rapid response team assessments [17].

To measure patient- and process-related outcomes, we used a wide variety of parameters, ranging from the standardized measurement tools that were validated for these rapid response team assessments to the times until interventions and general outcome data. Moreover, the environment and organization of the rapid response teams we assessed are in line with those in a multi-center trial that established a standard deployment of the rapid response teams which reduced in-hospital morality rates [3]. In our study, this organizational structure was considered efficient, based on the high number of calls that resulted in ICU admissions (60–70%) suggesting an effective afferent limb (detection), and on the relative fast reaction time as a parameter for the efferent limb (response) [18].

The absence of significant differences in outcomes between teams led by APPs and teams led by medical residents suggests that APPs are suitable alternatives for medical residents in leading rapid response teams. This finding is also supported by the higher number of calls handled per FTE by the APPs compared to the number of calls handled per FTE of medical residents of the ICU: one FTE APP handled approximately 2.5 times more calls than one FTE medical residents. An explanation for this substantial difference might be the continuity of care that APPs provide. This continuity is established by the APPs' continuous coverage in most shifts together with their presence alongside the residents. This continuity of care probably also explains the observation that APPs more easily decide to respond to the calls than the rotating medical residents, who regard these calls as stressful events (personal communication).

The outcomes of this study are in line with the outcomes of the few previously published studies, although the settings of those studies were different, which makes a sound comparison is difficult. One study showed no differences in quality after the introduction of a nurse practitioner as leader of a rapid response team [12]. The study compared a situation with and without a rapid response team, but in the group of patients treated by a rapid response team, the nurse practitioner was not always available, and deployment restrictions for the rapid response team were in place. Additionally, another publication reported an shorter time until admission to the ICU in a APP-led rapid response team [11]. Despite this improvement, however, the time span was much longer than the 4 hours suggested to be adequate by the MAELOR tool. Also our results are in line with the findings of a recent retrospective single-center analysis. This study compared an APP-led rapid response team with a resident-led rapid response team. This study found no differences between the groups except for a shorter in-hospital stay of the patients visited by the APP-led group but after propensity matching [13]. The study mainly focused on patient outcomes and less on process outcomes.

In accordance with this study, there is growing evidence that APPs are a valuable substitute for a physicians as leaders of a rapid response team.

Although the implementation of an RRT in this study is the general method used in the Netherlands, comparability between countries and healthcare systems remains cumbersome

because different RRT models that are used [3]. Apparently, so far, an optimal composition and implementation of a rapid response team has not yet been established [8, 19]. This fact and the lack of validated measurement tools except for the MAELOR tool probably explains why the literature reports different success rates and struggles with aligning outcome data when reporting on the performance of rapid response teams [3, 8, 13, 20, 21].

The main limitations of the present study are the retrospective design and the single-center setting. Before extrapolating our results to other hospitals, the local situation should be considered. Another limitation is that patients might have been missed if they did not fit our query, or were not registered in the database. However, we know that the database is used consistently and that registration is a central part of the workflow for all APPs and medical residents. It is therefore unlikely that selection is a major bias of this study. Another important limitation is that the patient- and process-related outcome measures were chosen arbitrarily, although they are clinically relevant. When focusing on time before intubation for example, there is a difference between the groups in time from admission to intubation without reaching significance. This is probably caused by the fact that there are a lot of oxygen therapies available which can be applied as initial treatment and if the included sample size is too small, significant differences in patient-related outcomes are difficult to detect. In addition, the outcomes are affected by many other variables related to the patient, the pathology, and the organization of the ward and the ICU. Especially, an ICU nurse may have a substantial influence on the rapid response team's performance. Necessary critical steps such as oxygen administration or positioning of the patient to enable adequate breathing are steps often overlooked by junior clinicians. Even simple treatment recommendations provided by the nurse, can be very valuable for junior clinicians. These steps are often independently covered by the accompanying ICU nurse.

Regarding the use of the MAELOR tool, this tool was validated to evaluate the performance of a rapid response team. However, the acquaintance of researchers with this tool is limited and therefore its use to compare between studies is limited as well. In addition, the added value of APPs might have been underestimated because certain benefits of APPs might have been missed, such as patient satisfaction, communication skills, team guidance, situational awareness, and other non-technical skills. This same issue has been addressed in the literature before [22]. Prospective cohort studies are therefore needed to confirm the outcomes of the present study and to assess the potential additional benefits of APP-led rapid response teams.

## Conclusion

In this observational retrospective single-center study on process- and patient-related outcome parameters of RRTs led by APPs and MRs, we have shown that APPs perform at least as well as MRs in leading a rapid response team. As the performance of rapid response teams is influenced by the organization of healthcare systems prospective studies in other institutions are needed to confirm our findings.

## Supporting information

**S1 Data.**
(XLSX)

## Author Contributions

**Conceptualization:** Ashley J. R. De Bie, Alexander J. G. H. Bindels, Peter H. J. van der Voort.

**Data curation:** Herman G. Kreeftenberg, Ashley J. R. De Bie, Eveline H. J. Mestrom.

**Formal analysis:** Herman G. Kreeftenberg, Ashley J. R. De Bie, Eveline H. J. Mestrom.

**Investigation:** Herman G. Kreeftenberg, Ashley J. R. De Bie, Peter H. J. van der Voort.

**Methodology:** Alexander J. G. H. Bindels, Peter H. J. van der Voort.

**Project administration:** Herman G. Kreeftenberg, Ashley J. R. De Bie, Eveline H. J. Mestrom.

**Resources:** Herman G. Kreeftenberg.

**Supervision:** Herman G. Kreeftenberg, Alexander J. G. H. Bindels, Peter H. J. van der Voort.

**Validation:** Ashley J. R. De Bie, Eveline H. J. Mestrom.

**Writing – original draft:** Herman G. Kreeftenberg.

**Writing – review & editing:** Herman G. Kreeftenberg, Ashley J. R. De Bie, Eveline H. J. Mestrom, Alexander J. G. H. Bindels, Peter H. J. van der Voort.

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
