## [Decision Letter · Decision Letter 0]

19 May 2022

PONE-D-21-35288The Advanced Practice Provider compared to the Medical Resident as the Leader of a Critical Care Outreach Team, a 12 month retrospective analysis.PLOS ONE

Dear Dr. Kreeftenberg,

Thank you for submitting your manuscript to PLOS ONE. After careful consideration, we feel that it has merit but does not fully meet PLOS ONE’s publication criteria as it currently stands. Therefore, we invite you to submit a revised version of the manuscript that addresses the points raised during the review process. Please see the reviewers' comments below. Please address in particular each of the reviewers' comments regarding clarifying the limitations of this study. Please submit your revised manuscript by Jul 01 2022 11:59PM. If you will need more time than this to complete your revisions, please reply to this message or contact the journal office at plosone@plos.org. Please include the following items when submitting your revised manuscript:A rebuttal letter that responds to each point raised by the academic editor and reviewer(s). You should upload this letter as a separate file labeled 'Response to Reviewers'.A marked-up copy of your manuscript that highlights changes made to the original version. You should upload this as a separate file labeled 'Revised Manuscript with Track Changes'.An unmarked version of your revised paper without tracked changes. You should upload this as a separate file labeled 'Manuscript'.

We look forward to receiving your revised manuscript.

Kind regards,

Hugh Cowley

Senior Editor

PLOS ONE

Journal Requirements:

2. "In your Data Availability statement, you have not specified where the minimal data set underlying the results described in your manuscript can be found. PLOS defines a study's minimal data set as the underlying data used to reach the conclusions drawn in the manuscript and any additional data required to replicate the reported study findings in their entirety. All PLOS journals require that the minimal data set be made fully available. For more information about our data policy, please see http://journals.plos.org/plosone/s/data-availability.

Reviewers' comments:

Reviewer's Responses to Questions

**Comments to the Author**

1. Is the manuscript technically sound, and do the data support the conclusions?

Reviewer #1: Yes

Reviewer #2: Yes

2. Has the statistical analysis been performed appropriately and rigorously? 

Reviewer #1: I Don't Know

Reviewer #2: Yes

3. Have the authors made all data underlying the findings in their manuscript fully available?

Reviewer #1: No

Reviewer #2: Yes

4. Is the manuscript presented in an intelligible fashion and written in standard English?

Reviewer #1: No

Reviewer #2: Yes

5. Review Comments to the Author

Reviewer #1: The authors present a retrospective observational study from a single tertiary centre in the Netherlands. They have two types of team leaders for rapid response calls, junior residents compared to physicians assistants or other trained staff. They report patient outcomes following these emergencies as well as some care access measures such as time to central line placement. Both groups of patients seem to have similar outcomes, although the patient numbers are very small for this type of study.

The study is largely well written. It is an important topic and it is good to see academic pilot exploration of the concept. It will be a good addition to the international literature on critical care delivery models in hospitals.

I would like to see data regarding the levels of training and experience of the two groups of providers (particularly the non-medical team leaders). When other sites consider rolling-out this staffing model, it will be important for them to understand that the response team is led by people who work full time in the ICU and have done for years. This is not likely to be the same resource available to rural centres, limiting the generalisability of the study, hence the importance of stating the setting of this study. (Also add a generalisability comment in your discussion).

I have a few minor suggestions for improvement of the manuscript:

# I recommend adding the time frame of your study to the abstract and methods.

# Data are pleural, this should be changed throughout the manuscript.

# The results should start with a participant flow diagram or statement. It is not clear if all rapid response team calls were included in your study at the moment.

# There is a sentence/section referring to the FTE of response by various healthcare workers to calls - this can be deleted, the gross numbers and percentages are enough. I think you are trying to convey levels of experience in these comments, which would be better dealt with another way.

# I recommend light editing throughout the manuscript by a native English speaker - the manuscript is well written but there are minor phrase and spelling issues throughout. eg "de" instead of "the"

# Table 1 should start with the numbers of patients in each column (n=xx)

# A lot of your narrative paragraphs in your results repeat data in Table 2 and could be deleted.

# Whilst you didn't reach statistical significance, your physicians assistants seem to be slower at initiating procedures in your ICU. This might be worth a comment. Some of the times are an hour or so longer - you seem not to have an adequate sample size if this doesn't reach statistical significance.

# if word counts are an issue, almost all sections could be more concise.

Reviewer #2: This paper compares rapid responses of teams led by a medical resident and by an advance practice provider. It is an interesting approach to broaden and valide skills of rapid response team leaders, based on the MAELOR tool. It does not however consider the role of the ICU nurse who is also part of the team, nor of the prior experiences of the leaders. These should be mentioned in the limitations. The accompanying nurse can play a major role in ensuring the quality of patient management: it is common for junior doctors to ask experienced nurses for their opinions and support, for example, and not considering them at all in the team's performance seems a bit of an oversight.

Overall, the presented analysis seems appropriate and the results are discussed with pertinence. It is interesting to consider making RRT leader specialists, since medical residents can have varying levels of expertise. It will be however important to ensure that medical residents continue to be exposed to and train for emergencies during their postgraduate training.

The are minor revisions needed for english (including : line 152, experienceS; line 161, no plural for advice; line 205, what is the word "en" ; line 216, correct continues for continuous; the paragraph 274-279 needs revision: the first sentence has no verb, the third is difficult to understand). The section in lines 263-266 are a repetition of the information ind lines 247-250.

6. PLOS authors have the option to publish the peer review history of their article (what does this mean?). If published, this will include your full peer review and any attached files.

Reviewer #1: No

Reviewer #2: **Yes: **Katherine Blondon

---

## [Author Response · Author response to Decision Letter 0]

15 Jun 2022

Dear Editor,

We thank the editor for the opportunity to revise our manuscript. We would like to offer the revised manuscript: Advanced Practice Providers versus Medical Residents as Leaders of Rapid Response Teams: a 12-month retrospective analysis to the editor and the reviewers for reconsideration.

We changed the title and used the designation rapid response team instead of critical care outreach team because this is the designation used throughout the text.

To address the requirements of the Journal we consulted a scientific English writer for proper scientific writing. Regarding the other remarks:

1. We addressed the data Availability Statement by adding our revised database to the ‘supplementary materials’. We deleted columns which could make the APPs, residents or patients traceable to make the database privacy proof.

2. We addressed the remark about style by changing the headings to the required fonts and by changing the file naming of the figure.

3. We addressed the contributorship by adding the statement that all authors contributed equally to this work. There was no difference in effort. If different symbols in the author list are still required we will be happy to change this. We were not able to differ between the authors based on level of contribution because it was a team effort.

The several valuable comments of the reviewers were addressed in the uploaded file named: rebuttal. 

For thoroughness we also copied the answers below.

Kind regards,

Herman Kreeftenberg

Rebuttal:

We thank de reviewers for their valuable feedback on the manuscript and we consider the requested changes as very helpful. To address the language and scientific writing we sent the manuscript to a native English scientific writer for improvement.

Comments reviewer #1:

1. I would like to see data regarding the levels of training and experience of the two groups of providers (particularly the non-medical team leaders). When other sites consider rolling-out this staffing model, it will be important for them to understand that the response team is led by people who work full time in the ICU and have done for years. This is not likely to be the same resource available to rural centres, limiting the generalisability of the study, hence the importance of stating the setting of this study. (Also add a generalisability comment in your discussion).

Answer: We added the experience of the APPs to the manuscript. We agree that experience is definitely a factor which is important to consider. What also has to be kept in mind that The APPs in this study also evolved from clinicians with very limited capabilities to valuable clinicians. We think that this reflects a dynamic environment which develops itself over years. The new APPs do not have the experience yet, but tend to learn very fast through guidance form the more experienced APPs. In this way this staffing model could also be applicable to healthcare organizations in rural areas. When deploying more APPs, clinical experience will develop faster and easier. It is probably the initial organizational setup which is most difficult. In addition, we experienced a transfer of nurses to become APPs. These nurses are often a selection which has an urge to develop and explore other professions. Because we offered them the opportunity to become an APP, these people continued to participate in our team. 

2. I recommend adding the time frame of your study to the abstract and methods.

Answer: We added the timeframe: the pre-COVID period 2017-2018. We chose this period because we think this does better reflect the ordinary duties of the APP and MR than a period during the COVID pandemic.

3. Data are pleural, this should be changed throughout the manuscript.

Answer: this has been corrected together with the language and style by editing of the manuscript by a native scientific writer. 

4.The results should start with a participant flow diagram or statement. It is not clear if all rapid response team calls were included in your study at the moment.

Answer: We added this statement at the beginning of the results section. Additionally, we explained why the number of intubations is less than the total number of patients: not all patients received this treatment. We therefore, did not delete all the text after the tables, because we considered it an explanation for the fact that we were not able to retrieve all data on every parameter.

5. There is a sentence/section referring to the FTE of response by various healthcare workers to calls - this can be deleted, the gross numbers and percentages are enough. I think you are trying to convey levels of experience in these comments, which would be better dealt with another way.

Answer: We deleted this section and made a general remark. We did considered this item worth mentioning because we experience some hesitation by the MR to confront critical situations on the normal wards, where the APP seems to consider these situations a challenge.

6. I recommend light editing throughout the manuscript by a native English speaker - the manuscript is well written but there are minor phrase and spelling issues throughout. eg "de" instead of "the"

Answer: We involved a native scientific writer for editing. We think the language and the manuscript has improved considerably.

7. Table 1 should start with the numbers of patients in each column (n=xx)

Answer: We added the number of patients to table 1.

8. A lot of your narrative paragraphs in your results repeat data in Table 2 and could be deleted.

Answer: we deleted a few repetitive parts of the text (as also mentioned by reviewer 2). We left some of the parts describing the results of table 2 unchanged because of the explanatory additional information in it. 

9. Whilst you didn't reach statistical significance, your physicians assistants seem to be slower at initiating procedures in your ICU. This might be worth a comment. Some of the times are an hour or so longer - you seem not to have an adequate sample size if this doesn't reach statistical significance.

Answer: We agree with the reviewer, and described this weakness in the limitation section of the manuscript. The relative small number of calls and the additional smaller number of intubations indicate a too small sample size. We added these parameters however, to explore additional parameter besides the raw patient outcome data which are almost always in line with the quality standards in a adequately organized healthcare organization and they do not often point to the favorability of one profession.

10. if word counts are an issue, almost all sections could be more concise.

Answer: Parts of the text were deleted and the suggestions of the scientific writer made the manuscript more concise.

Comments reviewer #2: 

This paper compares rapid responses of teams led by a medical resident and by an advance practice provider. It is an interesting approach to broaden and valide skills of rapid response team leaders, based on the MAELOR tool. 

1. It does not however consider the role of the ICU nurse who is also part of the team, nor of the prior experiences of the leaders. These should be mentioned in the limitations. The accompanying nurse can play a major role in ensuring the quality of patient management: it is common for junior doctors to ask experienced nurses for their opinions and support, for example, and not considering them at all in the team's performance seems a bit of an oversight.

Answer: We thank the reviewer for addressing this topic. We also value the important additional support and recommendations of the nurse and therefore, added this to the limitations section. In addition, to the other limitations this is another reason why the measurement of separate professions in a team is difficult. Mostly the entire team performance is measured.

2. Overall, the presented analysis seems appropriate and the results are discussed with pertinence. It is interesting to consider making RRT leader specialists, since medical residents can have varying levels of expertise. It will be however important to ensure that medical residents continue to be exposed to and train for emergencies during their postgraduate training.

Answer: We agree with this comment. This is the reason why we have a mixed team with a limited number of APPs although they are considered experienced and valuable for quality and continuity of critical care medicine.

3. There are minor revisions needed for English (including : line 152, experienceS; line 161, no plural for advice; line 205, what is the word "en" ; line 216, correct continues for continuous; the paragraph 274-279 needs revision: the first sentence has no verb, the third is difficult to understand). The section in lines 263-266 are a repetition of the information in lines 247-250.

Answer: We addressed this remark by consulting a native English speaking scientific writer, who offered suggestions for the improvement of the language and style of the manuscript. We think this led to a considerable improvement and a better readable manuscript.

---

## [Decision Letter · Decision Letter 1]

25 Jul 2022

PONE-D-21-35288R1Advanced Practice Providers versus Medical Residents as Leaders of Rapid Response Teams: a 12-month retrospective analysis.PLOS ONE

Dear Dr. Kreeftenberg,

Thank you for submitting your revised manuscript to PLOS ONE. After careful consideration, we feel that it has merit but does not fully meet PLOS ONE’s publication criteria as it currently stands. Therefore, we invite you to submit a revised version of the manuscript that addresses the points raised during the review process.

There remain a small number of changes suggested by reviewer 2 (see their comments below). Regarding the clarity of figure 1, the resolution is not a concern, but the contrast between the text and the background colour of the boxes (the red in particular) is poor. I would recommend choosing a different colour combination with better contrast, and also one that is more accessible to readers with red-green colour blindness. There are plenty of resources online for choosing a colour palette which is clear and accessible for all readers.

We look forward to receiving your revised manuscript.

Kind regards,

Joseph Donlan

Editorial Office

PLOS ONE

Journal Requirements:

Reviewers' comments:

Reviewer's Responses to Questions

**Comments to the Author**

1. If the authors have adequately addressed your comments raised in a previous round of review and you feel that this manuscript is now acceptable for publication, you may indicate that here to bypass the “Comments to the Author” section, enter your conflict of interest statement in the “Confidential to Editor” section, and submit your "Accept" recommendation.

Reviewer #2: (No Response)

2. Is the manuscript technically sound, and do the data support the conclusions?

Reviewer #2: Yes

3. Has the statistical analysis been performed appropriately and rigorously? 

Reviewer #2: Yes

4. Have the authors made all data underlying the findings in their manuscript fully available?

Reviewer #2: Yes

5. Is the manuscript presented in an intelligible fashion and written in standard English?

Reviewer #2: Yes

6. Review Comments to the Author

Reviewer #2: This version of the manuscript is much improved, both in content and in the use of english! My concerns have been addressed.

On page 10, line 208, I would suggest using "were" rather than "could be" for the categorization in 3 groups, because that is what you have done!

Finally, Figure 1 needs to be revised, I cannot read any of the information in the color boxes (problem with the contrast of colors, because the colors are very dark, as well as with the resolution of the text).

7. PLOS authors have the option to publish the peer review history of their article (what does this mean?). If published, this will include your full peer review and any attached files.

Reviewer #2: **Yes: **Katherine Blondon

---

## [Author Response · Author response to Decision Letter 1]

28 Jul 2022

Rebuttal second review:

We thank de reviewers for their latest valuable feedback on the manuscript and we consider all requested changes as very helpful. 

Comments reviewer #2:

1. Page 10, line 281: “were” instead of “could be”

Answer: Changed

2. Revise figure 1.

Answer: We want to thank the reviewer for this suggestion. We revised the figure and have improved the contrast between the positive (black, white text) and the negative (grey, black text) outcomes of the MAELOR tool. We also improved the directions of the flows in the flowchart.

---

## [Editor Report · Decision Letter 2]

4 Aug 2022

Advanced Practice Providers versus Medical Residents as Leaders of Rapid Response Teams: a 12-month retrospective analysis.

PONE-D-21-35288R2

Dear Dr. Kreeftenberg,

We’re pleased to inform you that your manuscript has been judged scientifically suitable for publication and will be formally accepted for publication once it meets all outstanding technical requirements.

Kind regards,

James Mockridge

Staff Editor

PLOS ONE

Additional Editor Comments:

In the Ethics section of the Methods, please amend your current ethics statement to include the full name of the ethics committee/institutional review board(s) that approved your specific study.

---

## [Editor Report · Acceptance letter]

11 Aug 2022

PONE-D-21-35288R2 

Advanced Practice Providers versus Medical Residents as Leaders of Rapid Response Teams: a 12-month retrospective analysis. 

Dear Dr. Kreeftenberg:

I'm pleased to inform you that your manuscript has been deemed suitable for publication in PLOS ONE. Congratulations! Your manuscript is now with our production department. 

Kind regards, 

on behalf of

Dr Joseph Donlan 

Staff Editor

PLOS ONE